# Effect of mitoTEMPO on Redox Reactions in Different Body Compartments upon Endotoxemia in Rats

**DOI:** 10.3390/biom13050794

**Published:** 2023-05-05

**Authors:** Adelheid Weidinger, Andras T. Meszaros, Sergiu Dumitrescu, Andrey V. Kozlov

**Affiliations:** 1Ludwig Boltzmann Institute for Traumatology, The Research Center in Cooperation with AUVA, 1200 Vienna, Austria; 2Department of Visceral, Transplant and Thoracic Surgery, Medical University of Innsbruck, 6020 Innsbruck, Austria

**Keywords:** reactive oxygen species, systemic inflammatory response syndrome, cytokines, mitochondria-targeted antioxidants, bronchoalveolar system, peritoneal lavage, mitoTEMPO

## Abstract

Mitochondrial ROS (mitoROS) control many reactions in cells. Biological effects of mitoROS *in vivo* can be investigated by modulation via mitochondria-targeted antioxidants (mtAOX, mitoTEMPO). The aim of this study was to determine how mitoROS influence redox reactions in different body compartments in a rat model of endotoxemia. We induced inflammatory response by lipopolysaccharide (LPS) injection and analyzed effects of mitoTEMPO in blood, abdominal cavity, bronchoalveolar space, and liver tissue. MitoTEMPO decreased the liver damage marker aspartate aminotransferase; however, it neither influenced the release of cytokines (e.g., tumor necrosis factor, IL-4) nor decreased ROS generation by immune cells in the compartments examined. In contrast, *ex vivo* mitoTEMPO treatment substantially reduced ROS generation. Examination of liver tissue revealed several redox paramagnetic centers sensitive to *in vivo* LPS and mitoTEMPO treatment and high levels of nitric oxide (NO) in response to LPS. NO levels in blood were lower than in liver, and were decreased by *in vivo* mitoTEMPO treatment. Our data suggest that (i) inflammatory mediators are not likely to directly contribute to ROS-mediated liver damage and (ii) mitoTEMPO is more likely to affect the redox status of liver cells reflected in a redox change of paramagnetic molecules. Further studies are necessary to understand these mechanisms.

## 1. Introduction

Systemic inflammation and consequent systemic inflammatory response syndrome (SIRS) is a complex cascade of pro- and anti-inflammatory processes. It is associated with release of inflammatory mediators such as tumor necrosis factor (TNF)-α and interleukins (IL-1, IL-6, IL-4, MCP1 and others) [1] as well as by excessive generation of nitric oxide (NO) by inducible nitric oxide synthase (iNOS), thought to be a mediator of distant organ damage [2]. The exact mechanism of cellular damage is not fully clarified, although the theory of a deleterious interplay of inflammatory mediators complemented by immune cell cascade is widely accepted [1]. The question remains of the extent to which immune cells and their products play a causative role in the distant organ damage.

The inflammatory response is often accompanied by multiple organ dysfunction syndrome (MODS), which is often associated with excessive generation of reactive oxygen and nitrogen species (RONS). It has been shown that this process is orchestrated by mitochondrial ROS (mitoROS), mainly superoxide (O2^•−^), a primary ROS generated by the electron transfer system in the mitochondrial inner membrane [3]. However, it is very difficult to dissect direct effects of ROS *in vivo* due to their very short lifetime. A common and precise way to access these biological effects *in vivo* is the examination of effects of so-called mitochondria-targeted antioxidants (mtAOX). These molecules comprise two structural parts. The first is positively charged and hydrophobic. This part is considered as chemically inert, and brings the entire molecule to the negatively charged mitochondrial matrix. The second part has specific ROS-scavenging moieties [4]. TEMPO is the most often used antioxidant part of the molecule.

Generally, the major reason for increased generation of mitochondrial ROS is a shift in the redox equilibrium within the cells. These disturbances are often accompanied by changes in intracellular redox active centers and deviation of electrons from their physiological pathways (redox shuttles) to one-electron reduction of oxygen-yielding superoxide radicals [5]. Such redox-motivated electron transfer appears in the mitochondrial electron transport chain (ETC) or electron transport system linked to cytochrome P450 in the endoplasmic reticulum. One-electron reduction of oxygen appears under pathological circumstances such as hypoxia or inflammation. The major sites of leakage of electrons in mitochondria are Complex I and III of the electron transfer system [6]. Another important component of the respiratory chain is ubiquinone (Q10), which exerts both pro- and antioxidant capacities [7]. A number of these centers containing enzymes of mitochondria, such as P450, Q10, and iron, can be detected in a specific redox state by electron paramagnetic resonance spectroscopy (EPR).

Spatially, there are two general sources of ROS that are considered potentially damaging for tissues. Extracellular ROS are mainly generated by immune cells [8], while intracellular ROS production leads to cell damage from intracellular sources [2]. Thus, it can be expected that the antioxidant mitoTEMPO acts both in intra- and extra-cellular environments owing to the almost ubiquitous presence of mitochondria.

To this end, the effects of mtAOX on ROS generation and on various redox processes were evaluated in a lipopolysaccharide (LPS)-induced rodent SIRS model. LPS induces acute systemic inflammatory response in rodents [9] and humans [10]. This model mirrors certain aspects of septic shock in humans, though the correlation between rodent LPS models and clinical septic shock is poor [11]. Nonetheless, this model has provided the majority of mechanistic insights into systemic inflammation [10], such as the release of cytokines [12], induction of apoptosis [13], signal transduction [14], and others.

In the present study, we set out to characterize ROS generation in three major pathologically relevant sites of oxidative stress: (1) in the intravascular compartment (blood), (2) in the intraabdominal compartment, and (3) in the bronchoalveolar system. To the best of our knowledge, there are no studies on the effects of mitoTEMPO in both body compartments and in a reference tissue (liver) susceptible to inflammatory damage.

## 2. Materials and Methods

### 2.1. Chemicals

All chemicals were obtained from Sigma-Aldrich (St. Louis, MO, USA) unless otherwise noted.

### 2.2. Animals

The *in vivo* experiments were performed on male Sprague Dawley rats (250–300 g/390–540 g; Animal Research Laboratories, Himberg, Austria/Charles River, Germany) which were kept under controlled standard animal housing conditions at least for 7 days prior to usage in experiments with free access to standard laboratory rodent food and water. The rats with high weight were used in experiments first; small animals were kept approximately 2 weeks longer in the animal research facility of the institute. All interventions were conducted in compliance with the Guide for the Care and Use of Laboratory Animals published by the National Institute of Health with approval from the Animal Protocol Review Board of the city government of Vienna, Austria (no. M58003956-2011-9, no. 815758/2014/16). To prevent unnecessary pain, Buprenorphin (Richter Pharma AG, Wels, Austria, 0.05 mg/kg body weight) was injected subcutaneously at the time of the LPS treatment and 10 h thereafter. At the end of experiments, rats were anesthetized by inhalation of a mixture of 3% isoflurane and oxygen (Vapor 2000, Dräger, Austria).

### 2.3. Lipopolysaccharide and mtAOX Treatment

Rats were injected with lipopolysaccharide (LPS) from Escherichia coli serotype 026:B6 (activity ≥ 500,000 EU/mg) at a dose of 2.5 mg/kg body weight dissolved in saline (Fresenius Kabi, Graz, Austria). Samples were collected at 16 h after LPS injection. In a separate set of experiments (Figure 1), animals were divided into four groups. All rats were injected with the same dose of LPS (Escherichia coli serotype 026:B6, 8 mg/kg body weight, activity ≥ 10,000 EU/mg) dissolved in saline. Control animals were injected with saline only. Treatment with mitoTEMPO (50 nmol/kg) was administered intraperitoneally at 1 h before LPS treatment and 11 h after LPS treatment. The LPS solution was vortexed for 1 min and sonicated for 30 s before application, then injected in the penis vein under isoflurane anesthesia (Vapor 2000, Dräger, Austria) in volumes ranging from 0.5 to 0.75 mL.

### 2.4. Sampling of Blood and Liver Tissue

After a small skin cut was made, the left femoral artery was dissected and catheterized using a 24 GA i.v. cannula (BD Neoflon, Becton Dickinson Infusion Therapy AB, Helsingborg, Sweden). Next, 10–12 mL of whole blood was collected in a 50 mL Falcon tube prefilled with 200 µL of sodium heparin (1000 IU/mL, Gilvasan Pharma GmbH, Austria) for subsequent processing and for the in vitro part of the study.

For determination of mononitrosyl–hemoglobin complex (NO-Hb) levels, blood samples were taken into Minicollect tubes coated with lithium heparin (Greiner Bio-One GmbH, Kremsmünster, Austria). After centrifugation at 4 °C and 1600× *g* for 10 min, plasma was removed and the erythrocyte pellet was collected in 1 mL plastic syringes. Subsequently, erythrocyte pellets were shock-frozen in liquid nitrogen and stored at −80 °C until EPR measurement.

Following blood sampling, animals were euthanized by decapitation. Subsequently, the liver was excised and transferred immediately to a beaker filled with ice cold Ringer solution (Fresenius Kabi, Austria). After cooling, the liver was cut into small pieces on a Petri dish placed on ice. Tissue samples to a volume of 0.4 mL were filled in 1 mL plastic syringes, shock-frozen in liquid nitrogen, and stored at −80 °C for further measurements. Liver tissue for measurement of mitochondrial respiration was used freshly.

### 2.5. Measurement of Mitochondrial Respiration

Respiratory parameters of mitochondria were monitored using high-resolution respirometry (Oxygraph-2k, Oroboros Instruments, Innsbruck, Austria) as previously described [15]. Rat liver homogenate was incubated in buffer containing 105 mM KCl, 5 mM KH_2_PO_4_, 20 mM Tris-HCl, 0.5 mM EDTA, and 5 mg/mL fatty acid-free bovine serum albumin (pH 7.2, 37 °C). Respiration was stimulated by the addition of 5 mM glutamate and 5 mM malate. Transition to State-3 respiration was induced by the addition of 1 mM adenosine diphosphate. Oxygen consumption rates were obtained by calculating the negative time derivative of the measured oxygen concentration. The respiratory control ratio was calculated by dividing State 3 respiration by State 2 respiration.

### 2.6. Electron Paramagnetic Resonance Spectroscopy

EPR is the most appropriate method to detect redox active centers directly in untreated tissues. It can be used to determine redox-active iron-sulfur species [16], copper containing compounds [17], free radicals and ferrous ions [18], and nitric oxide [19]. A particular advantage of this method is that the analytic procedure can be performed directly in frozen tissue biopsies at liquid nitrogen temperature [19]. This ensures the absence of artefacts due to processing of tissues, such as homogenization, extraction etc.

EPR spectra were recorded at liquid nitrogen temperature (−196 °C) with a Magnettech MiniScope MS 200 EPR spectrometer (Magnettech Ltd., Berlin, Germany) at a modulation frequency of 100 kHz and microwave frequency of 9.429 GHz. The settings for NO-Hb spectra in blood samples were microwave power 8.3 mW and modulation amplitude 5 G. NO-Hb complexes were recorded at 3300 ± 200 G and quantified by double-integrating the EPR spectra. The settings for NO-Hb and Fe-NO complex in liver samples were microwave power 30 mW and modulation amplitude 6 G. Liver spectra were recorded at 3200 ± 500 G. The settings for p450 (g = 2.25), free radicals (g = 2.002), and succinate dehydrogenase (g = 1.94) were microwave power 1 mW and modulation amplitude 5 G). The intensity of the signals was recorded at 3200 ± 1000 G.

### 2.7. Analysis of Total Nitric Oxide

Total nitric oxide levels (NOx) were analyzed with Sievers 280i-NO Analyzer (General Electric Company, Boston, MA, USA) as previously described [20]. Plasma samples were injected through a septum into the glass vessel, where NO species were converted by a redox active reagent (VCl3) to NO(g). Subsequent reaction with ozone causing photon emission was detected as chemiluminescence intensity.

### 2.8. Sampling of Peritoneal and Bronchoalveolar Lavage

Bronchoalveolar lavage was collected as described elsewhere [21]. A 19-gauge needle hub was inserted into the trachea and 3 mL of ice-cold phosphate-buffered saline (PBS) and 10% fetal bovine serum (FBS) were administered and aspirated slowly through the needle hub. After transfer into a 15 mL tube (Greiner Bio-One, Kremsmünster, Austria), the fluid was kept on ice. The procedure was repeated four times.

For collection of peritoneal cells, we used a protocol adapted from [22]. Briefly, the peritoneum was exposed and 5 mL of ice-cold PBS with 3% FBS was injected into the peritoneal cavity through a needle. A gentle massage to the abdomen was applied to dislodge any attached cells into the PBS solution. The suspension was collected through a 22-gauge needle into a syringe and transferred into a 15 mL tube, then kept on ice. The above-described procedure was repeated five times.

### 2.9. Isolation of Cells

The suspension of bronchoalveolar and peritoneal cells was centrifuged at 400× *g* for 10 min at 4 °C, the supernatant was discarded, and the cell number was counted with a Cell-Dyn 3700 Hematology Analyzer (Abbott Laboratories, Lake Bluff, IL, USA). Cell counts in blood and in peritoneal and bronchoalveolar fluids showed mainly polymorphonuclear leukocytes (PMN, mainly neutrophil granulocytes; see Appendix A). After the blood count, blood samples were centrifuged at 300× *g* for 10 min at 4 °C, then the plasma and buffy coat were removed and the remaining blood pellet was incubated with a lysis buffer containing 168 mM NH4Cl, 10 mM KHCO3, and 973 μM EDTA for 10 min at 4 °C. Cells were then washed twice with 10 °C cold PBS and centrifuged at 400× *g* for 8 min before finally performing the counts. For the in vitro mtAOX treatment, cell suspensions were incubated at 37 °C for 45 min with mitoTEMPO (500 nM) or vehicle (NaCl).

### 2.10. Ex Vivo ROS Measurement

The H_2_O_2_ production in the cell suspension was assessed by N-acetyl-3,7-dihydroxy phenoxazine (Amplex Red, Life Technologies, Eugene, OR, USA). Amplex Red is a sensitive and chemically stable fluorogenic probe, and produces fluorescent resolufin with H_2_O_2_ in a horseradish peroxidase (HRP)-catalyzed oxidation with excitation/emission maxima at 563/587. The reaction stoichiometry of Amplex Red and H_2_O_2_ is 1:1 [21]. Using a fluorescence plate reader (POLARstar Omega 3MG, Labtech, Germany), white blood cells, bronchoalveolar lavage, and peritoneal lavage cells (15 × 10^3^ cells per well in Krebs buffer) were incubated in black 96-well microplates (Greiner Cellstar) at 37 °C with Amplex Red (10 μM) and HRP (0.2 U mL^−1^). The fluorescence intensity was recorded for 30 min with an excitation of 544 nm/emission of 590 nm and gain at 1200. The slope was calculated from a 10 min interval between the 5th and 15th minute of the measurement.

### 2.11. Statistics

Statistical evaluation of the experimental results was performed using GraphPad Prism(Version 9.4.1., GraphPad Software, Boston, MA, USA). The results are shown as the mean ± standard error of the mean (SEM). Statistical significance was evaluated by ANOVA followed by Post Hoc, Holm-Šídák’s multiple comparisons test unless indicated otherwise in the figure legends; n numbers are indicated in the figure legends. Statistical significance is indicated as follows: * *p* < 0.05; ** *p* < 0.01; *** *p* < 0.001; **** *p* < 0.0001.

## 3. Results

### 3.1. LPS Treatment Leads to Tissue Damage in an mtAOX-Dependent Way

As a first step, we validated our rat endotoxemia model (Figure 1).

By 16 h post-LPS challenge, an increase in aspartate aminotransferase AST (Figure 2a) could be observed. MitoTEMPO pretreatment, however, could ameliorate this parenchymal damage. According to the literature, an early peak of tumor necrosis factor (TNF)-α can be seen at 2 h post-LPS in the plasma [2]. In this model, *in vivo* treatment with mitoTEMPO neither influenced the release of early acute phase cytokines (TNF, IL-1) into the circulation (Figure 2b,c) nor substantially decreased the levels of cytokines such as IL-4 and MCP-1 linked to the activation of monocytes (Figure 1d,e). In addition, we found that mitochondrial function in the liver is affected by LPS challenge (Figure 1f). As with the AST release, addition of mitoTEMPO normalized the respiratory control ratio (Figure 1f).

### 3.2. In Vivo mitoTEMPO Treatment Does Not Influence Ex Vivo ROS Generation

Next, we examined the *ex vivo* ROS generation by immune cells isolated from peripheral blood, peritoneal fluid, and lavage from the bronchoalveolar system. However, we did not observe any change in ROS generation in response to *in vivo* mitoTEMPO treatment in any of the studied compartments (Figure 3a–c). Further, to test whether the capacity of cells to generate ROS is affected by mitoTEMPO at all, we activated cells *ex vivo* by phorbol 12-myristate 13-acetate (PMA); again, we did not find any difference in this case (Figure 3d–f). Treatment with mitoTEMPO did not influence the relative counts of immune cells in any of the three compartments (Appendix A). These data suggest that either ROS generation by immune cells is not regulated by mitoROS or mitoTEMPO did not affect immune cells *in vivo*.

### 3.3. Ex Vivo mitoTEMPO Application Reduces ROS Production

To investigate the direct effects of mitoTEMPO on ROS production of immune cells, cells were similarly extracted from all body compartments as described above. In this case, mitoTEMPO was applied *ex vivo*. In contrast to the *in vivo* treatment, mitoTEMPO substantially reduced the rate of ROS generation (Figure 4a–c). Interestingly, this inhibition disappeared when cells were additionally activated *ex vivo* by PMA, with the exception of bronchoalveolar lavage cells (Figure 4d–f).

### 3.4. EPR Reveals mitoTEMPO-Dependent Decrease in Free Radical and NO Signals

Investigation of redox-sensitive intracellular paramagnetic centers using EPR technique (Figure 5a) revealed that their redox state is responsive to both LPS and mitoTEMPO. Both substances increased the intensity of the signal coming from p450 (Figure 5b), while the free radical signal, which originates predominantly from mitochondrial Q10, was increased by LPS (Figure 5c), though this increase was diminished by mitoTEMPO (Figure 5c). The signal g = 1.94, which is usually attributed to succinate dehydrogenase of mitochondria, was increased by LPS but did not change in response to mitoTEMPO (Figure 5d). In LPS-treated rats, the concentration of mononitrosyl-hemoglobin-complexes (NO-Hb) determined in liver tissue (Figure 5f) was higher than in the blood (Figure 5e), supporting the assumption that liver is a (the) prominent NO source, eventually higher than the vasculature and the blood itself. Increased concentrations of NO-Hb were attenuated by mitoTEMPO (Figure 5e,f). Similar changes were observed with liver dinitrosyl iron complexes (Fe-NO) reporting intracellular NO levels (Figure 5g). In addition, NOx levels in plasma after LPS treatment, determined by ozone-chemiluminescence technology, were increased (Figure 5h). This increase could be reduced by mitoTEMPO treatment (Figure 5h).

## 4. Discussion

In the present study, we employed a rodent model, which is extensively described in the literature, to investigate systemic inflammation and consequent organ damage. Bacterial endotoxin (LPS) treatment is known to lead to a dose-dependent increase in mortality along with elevation of organ damage markers and nitric oxide levels, as well as substantial changes in plasma cytokine pattern [23,24]. It is generally accepted that an inflammatory activation of cytokines orchestrated by and acting through ROS play a central role in end-organ damage [25].

Mitochondria-targeted antioxidants such as mitoTEMPO are widely employed to elucidate functional aspects of ROS-dependent mechanisms. Because we have already investigated the effect of mitoTEMPO on control tissues in previous studies [2,26], we did not include this group here. Consequently, the conclusions are limited to comparisons between control and LPS on one hand, and on the other LPS and LPS with mitoTEMPO.

In the present study, mitoTEMPO treatment successfully mitigated the tissue damage to the liver parenchyma, as evidenced by AST levels. This is in line with previous publications [2]. We observed that the respiratory control ratio is increased in response to LPS and normalized upon addition of mitoTEMPO. It has already been reported that upon inflammatory conditions the mitochondrial respiratory function can respond by a decrease or an increase in the capacity of the electron transport chain [27]. In our previous studies on rodents, we observed an increase in the State 3 respiration [28], as we show here, while in experiments with peritonitis in pigs the respiratory activity was decreased [29]. The reason that mitochondrial function is upregulated in certain cases and downregulated in others is not completely clear. However, the normalization of mitochondrial function to its control values by mitoTEMPO suggests that the changes in mitochondrial function observed here reflect pathological changes in the liver and that these changes are mediated by mitoROS.

According to the literature, an early peak of TNF-α can be seen at 2 h post-LPS [2]. TNF-α and IL-1 are central cytokines in the acute systemic inflammatory response, on the one hand exerting direct effects on hepatocytes, inducing apoptosis and necrosis [30], and on the other hand sending later cytokines into action. Our data suggest that fast (2 h and earlier) release of TNF/IL-1 at the beginning of acute inflammatory response orchestrates the immune response rather than directly contributing to the liver damage. This may indeed be possible, as the first wave of TNF/IL-1 is released from membrane-bound pools of these inflammatory mediators and not from a pool which needs to be synthetized de novo. This does not preclude the fact that in the later second wave of their release, when they are upregulated on the genetic level, their levels will be sensitive to mitoTEMPO (Figure 6).

Assuming effective intramitochondrial radical scavenging properties of mtAOX, we can conclude the following based on the experimental data. Beneficial effects of mtAOX on hepatocellular injury by 16 h are not directly linked to TNF-α, as mtAOX prevented AST release despite no alterations in TNF levels. Either later cytokines in the cascade are ROS-sensitive, or another process is contributing to cellular damage.

To determine whether ROS production in different body compartments can be modulated by mitoTEMPO, we examined ROS generation by immune cells in the blood, peritoneal fluid, and lavage from the bronchoalveolar system. We predominantly obtained PMNs from the three body compartments. Considering the central role of circulating PMNs in this systemic inflammation model, which is not TNF-α-dependent, an explanation for the protective effects could be a direct effect of mtAOX on PMNs. Although these cells have only a few functional mitochondria, a growing body of evidence suggests important functions of this organelle in initiation and migration [31], and Dikalov [32] and Daiber [33] have previously reported a feed-forward loop between mtROS and nicotinamide adenine dinucleotide phosphate (NADPH) oxidase activity. Theoretically, by inhibiting mtROS less ROS is formed by NADPH oxidase, resulting in less parenchymal cellular damage. Thus, we designed an experiment to investigate a direct effect of mtAOX on PMNs. There was no difference between PMA-induced ROS production of treated and non-treated groups, implying no causative role of reduced oxidative bursts of circulating immune cells capable of eliciting oxidative damage in the target organ in this setting.

We failed to reproducibly determine ROS generation of controls, and analyzed only the cells obtained from animals treated with LPS. In contrast to our expectations, *in vivo* injection of mitoTEMPO did not influence the rate of ROS generation in any of the examined compartments. Interestingly, the *ex vivo* treatment of the same cells with mitoTEMPO substantially reduced ROS generation. We assume that mitoTEMPO is distributed inside the body within the tissues; however, at the time when PMNs generate ROS it is possible that mitoTEMPO may not be present in sufficiently high concentrations in the corresponding compartments. In addition, it is possible that these cells might not be completely activated, as external treatment with PMA induced an additional release of ROS. The ROS release induced by PMA was not sensitive to mitoTEMPO. These data suggest that effect of mitoTEMPO in terms of ROS release is not mediated by activated immune cells.

Our observation that mitoTEMPO attenuated ROS generation *ex vivo* when exposure occurred prior to PMA treatment suggests that the generation of extracellular ROS in all three compartments upon treatment with LPS is controlled by mitoROS. This may not be the case when cells are activated by PMA. Indeed, PMA may activate immune cells in a mitoROS-independent manner by directly activating protein kinase C (PKC), and subsequently NADPH oxidase. However, LPS is known to act via toll-like receptors (TLR), which trigger the activation of NADPH-oxidases over mitochondrial ROS [34].

When analyzing the EPR spectra of the liver we observed specific changes of redox sensitive paramagnetic centers, particularly those involved in mitochondrial ROS generation and scavenging. We observed an increase in the Q10 signal in response to LPS, which was attenuated by mitoTEMPO. Q10 is an antioxidant in its reduced state, and it is able to scavenge ROS to yield stable free radicals. The latter can be detected by EPR. *In vivo* treatment with mitoTEMPO reduced this signal to its normal levels. Induction of iNOS is a sign of ongoing inflammation, and it has previously been shown that iNOS is upregulated in our model [2]. In our experiments, we observed increased NO levels in liver tissue and blood. The concentration of NO in liver tissue was higher than in the blood, suggesting that NO is formed predominantly in the liver (and probably other tissues) and then released into blood. *In vivo* treatment with mitoTEMPO reduced the NO levels elevated by LPS.

Based on the data presented here, and according to the literature [9], it can be concluded that classical early pro-inflammatory cytokines (TNF-α, IL-6) alone are not direct mediators of tissue damage *in vivo*. MtAOX treatment prevented hepatocellular damage despite obvious elevation of TNF-α in the circulation. Our data suggest that inflammatory mediators orchestrate systemic immune response, rather than directly contributing to ROS-mediated liver damage. Because ROS can originate in extracellular and intracellular compartments, it is expected that the mitochondrial antioxidant mitoTEMPO should prevent ROS-induced organ damage. This is supported by the fact that mitochondrial ROS regulate the release of intracellular and extracellular ROS from other sources [3]. Consequently, mitoTEMPO influences the redox status of liver cells, which can contribute to its local beneficial effect, though it may not be sufficient to reduce the systemic ROS generation by immune cells. The latter is likely due to the major portion of mitoTEMPO being absorbed by tissues and only a small portion being absorbed by immune cells. This may be explained by the volume/absorption surface of the tissues, which is much higher than the surface of immune cells.

## Figures and Tables

**Figure 1 biomolecules-13-00794-f001:**
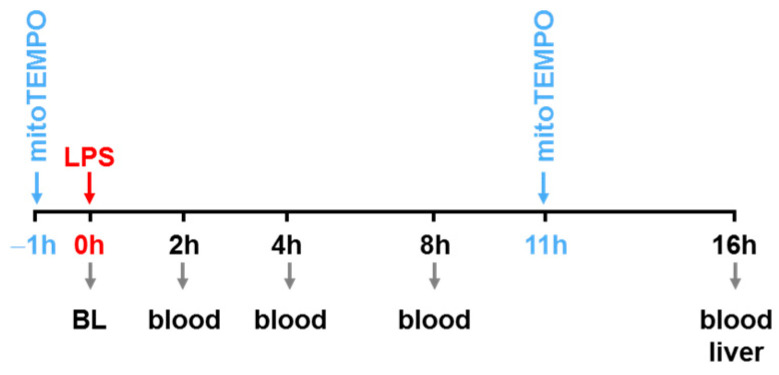
Experimental protocol of LPS and mitoTEMPO treatment followed by blood and tissue sampling. Treatment with mitoTEMPO was administered intraperitoneally at 1 h before LPS treatment and 11 h after LPS treatment. Blood samples were collected at 0/2/4/8/16 h and tissue samples at 16 h.

**Figure 2 biomolecules-13-00794-f002:**
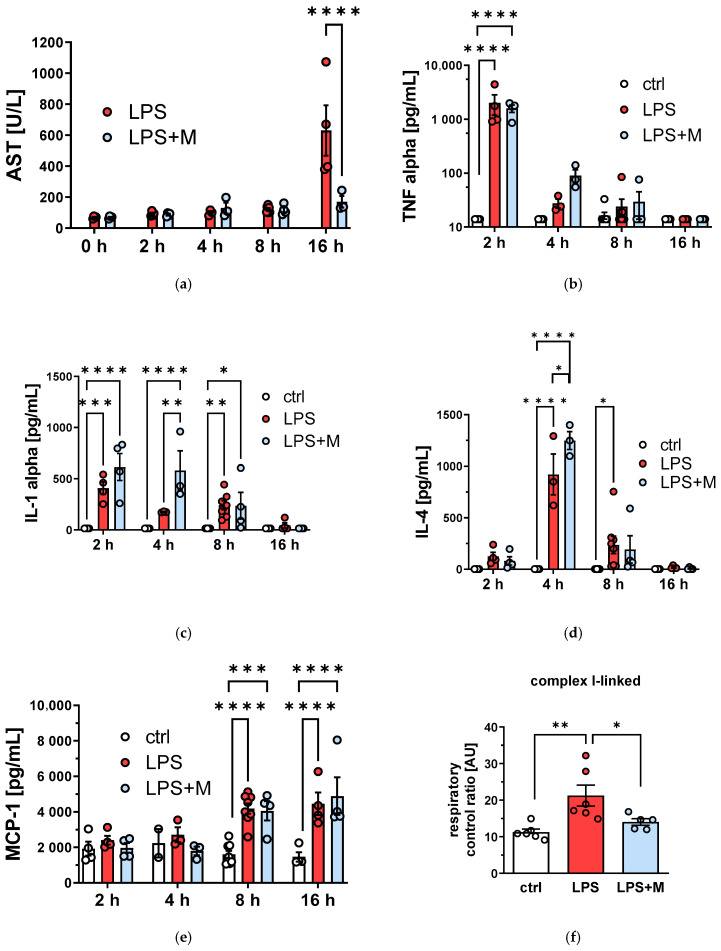
Effect of mitochondria-targeted antioxidant (mitoTEMPO, M) on liver function and humoral immune response after LPS treatment: (**a**) time course of AST (**b**), TNF-alpha (**c**), IL1-alpha (**d**) IL-4 (**e**), and MCP1. The animals were treated with 106 U/kg (approx. 2 mg/kg) LPS or saline (control group) and observed for up to 16 h. Samples were taken at 2/4/8/16 h. (**f**) Mitochondrial respiratory control ratio. Liver samples were taken at 16 h. The data are presented as mean ± SEM. Statistical evaluation was performed by ANOVA followed by Holm-Šídák’s multiple comparisons test. n = 3–8. Statistical significance is indicated as follows: * *p* < 0.05; ** *p* < 0.01; *** *p* < 0.001; **** *p* < 0.0001.

**Figure 3 biomolecules-13-00794-f003:**
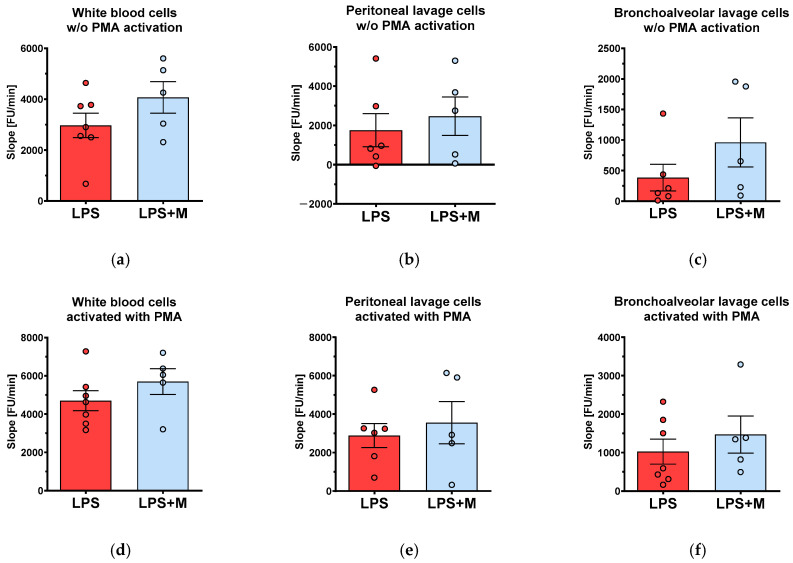
*In vivo* effects of the mitochondria-targeted antioxidant mitoTEMPO (M) on ROS generation in blood and bronchoalveolar lavage after LPS treatment: blood (**a**,**d**) and peritoneal (**b**,**e**), and bronchoalveolar (**c**,**f**) lavage were collected under standardized conditions as described in the methods section. ROS generation was measured in cells without activation with PMA (**a**–**c**) and after *ex vivo* activation with PMA (**d**–**f**). The data are presented as mean ± SEM. n = 5–7. Statistical evaluation was performed by ANOVA followed by Holm-Šídák’s multiple comparisons test.

**Figure 4 biomolecules-13-00794-f004:**
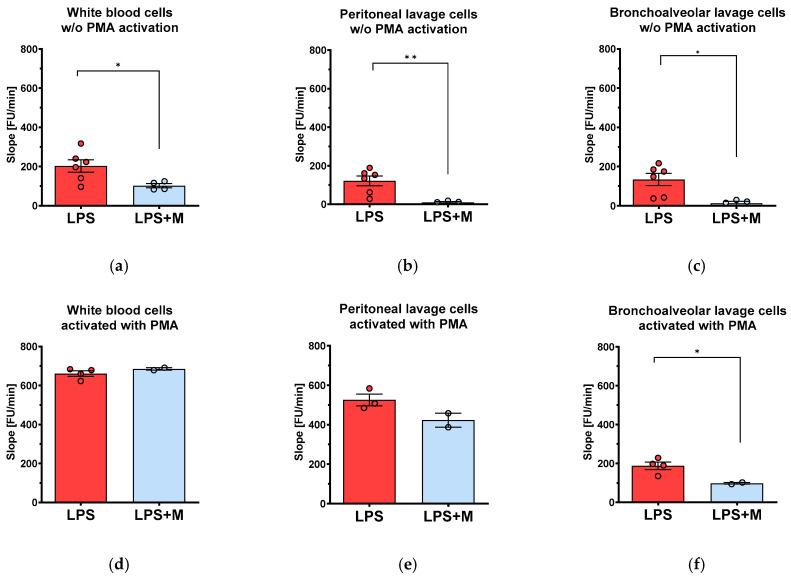
*Ex vivo* effects of mitochondria-targeted antioxidant mitoTEMPO (M) on ROS generation in blood and peritoneal and bronchoalveolar lavage exposed to LPS. Blood (**a**,**d**) and peritoneal (**b**,**e**), and bronchoalveolar (**c**,**f**) lavage were collected under standardized conditions as described in the methods section. ROS generation was measured in cells without activation with PMA (**a**–**c**) and after *ex vivo* activation with PMA (**d**–**f**). The data are presented as mean ± SEM. n = 2–6. Statistical evaluation was performed by ANOVA followed by Holm-Šídák’s multiple comparisons test. Statistical significance is indicated as follows: * *p* < 0.05; ** *p* < 0.01.

**Figure 5 biomolecules-13-00794-f005:**
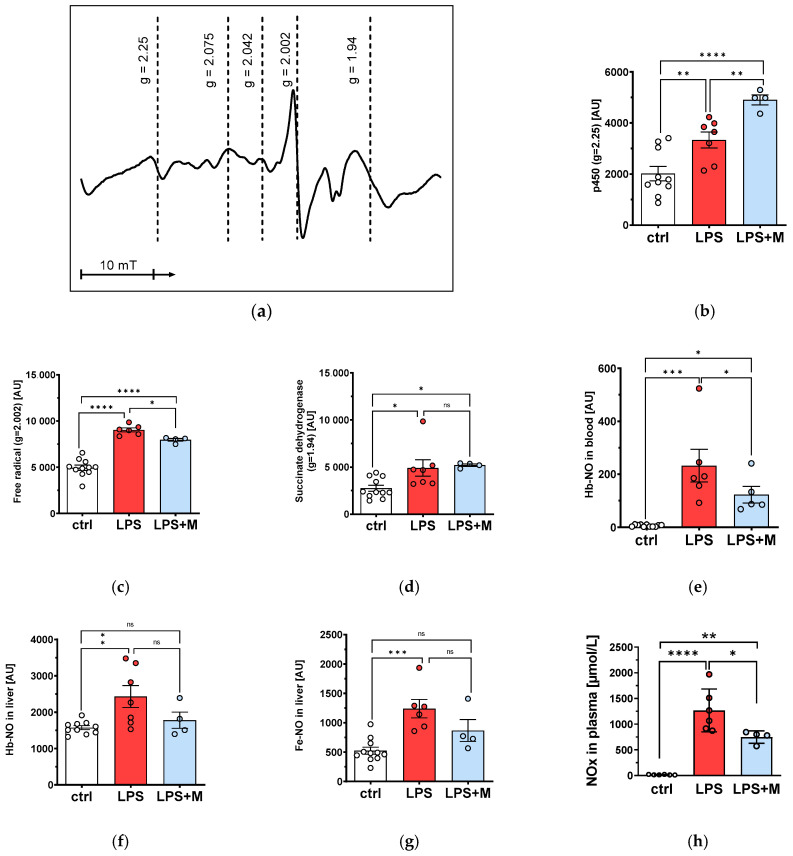
Redox state of paramagnetic centers: EPR spectrum of liver tissue (**a**); changes in the intensity of the cyt P450 signal (g = 2.25) (**b**); changes in the intensity of tissue free radical signal (g = 2.002) (**c**); changes in the intensity of succinate dehydrogenase signal (g = 1.94) (**d**); NO-Hb levels in circulating blood (**e**) and blood in liver vessels (**f**); Fe-NO levels in liver cells (**g**); NOx level in plasma (**h**). The data are presented as mean ± SEM. Statistical evaluation was performed by ANOVA followed by Holm-Šídák’s multiple comparisons test. n = 4–10. Statistical significance is indicated as follows: * *p* < 0.05; ** *p* < 0.01; *** *p* < 0.001; **** *p* < 0.0001.

**Figure 6 biomolecules-13-00794-f006:**
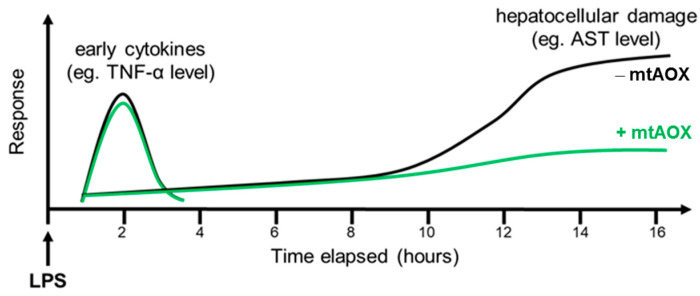
Schematic presentation of the time course of early cytokine response and hepatocellular damage after intravenous LPS challenge with and without mtAOX treatment in rats. LPS-induced hepatocellular damage, reflected by serum AST levels, is shown on the same timescale. AST can be seen rising slightly with the time following LPS injection, reaching significance compared to baseline levels at 8 h post-LPS. MtAOX treatment shows a strong reduction in circulating AST levels 16 h post-LPS, indicating a protective effect of mtAOX, and conversely a key role of mtROS in organ damage induced by endotoxemia.

## Data Availability

The data of this study are available on request from the corresponding author, [A.V.K.].

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
