# Peer review of "Effect of mitoTEMPO on Redox Reactions in Different Body Compartments upon Endotoxemia in Rats"

_biomolecules, 2023, doi:10.3390/biom13050794_

Round 1

Reviewer 1 Report

In their very well written paper, Weidinger et al. report how mitoROS affect redox reactions in different body compartments in their rat model of endotoxemia. They induced an inflammatory response by injection of lipopolysaccharide (LPS) and analyzed the effects of mitoTEMPO (mitochondria-targeted antioxidants) in blood, abdominal cavity, bronchoalveolar space, and liver tissue.

Based on the data presented, the authors conclude that classical early proinflammatory cytokines (TNF-α, IL-6) alone are not direct mediators of tissue damage in vivo. Treatment with MtAOX prevented hepatocellular injury despite a marked increase in TNF-α in the circulation. They suggest that inflammatory mediators orchestrate the systemic immune response and do not directly contribute to ROS-mediated liver injury. MitoTEMPO affects the redox status of liver cells, which may contribute to its beneficial effect, but still is not sufficient to reduce systemic ROS generation by immune cells. The latter is likely because most of MitoTEMPO is taken up by tissues and only a small fraction by immune cells. This may be explained by the volume/abortive surface area of the tissues, which is much larger than the surface area of the immune cells.

This well-conducted study and well-written manuscript should be published after minor issues commented by the reviewer are resolved and contribute interesting findings to the scientific community and research field.

minor remarks:

materials and methods: personally the reviewer would prefer for optimal transparency, to present here the animal experimentation ethic permit number. line 88/89

Result section:

Figure 1 b-d are missing in my download of the manuscript.

Figure 1a, “blood and organ sampling” together is a bit misleading, organs were harvested at the end of this period. Of course the reviewer understands what is meant, but maybe this could be optimized.

Results 3.4. line 186 and 187: the reviewer is convinced that in field of the authors everyone knows what EPR technique might be, but maybe not for everyone in general this is clear.

Discussion, Line 202

“murine model” would be a mouse model, in the manuscript a “rat model” or “rodent model” is described.

Author Response

Point-by-point Response to Reviewer's Comments

Reviewer#1

Reviewer#1: In their very well written paper, Weidinger et al. report how mitoROS affect redox reactions in different body compartments in their rat model of endotoxemia. They induced an inflammatory response by injection of lipopolysaccharide (LPS) and analyzed the effects of mitoTEMPO (mitochondria-targeted antioxidants) in blood, abdominal cavity, bronchoalveolar space, and liver tissue. Based on the data presented, the authors conclude that classical early proinflammatory cytokines (TNF-α, IL-6) alone are not direct mediators of tissue damage in vivo. Treatment with MtAOX prevented hepatocellular injury despite a marked increase in TNF-α in the circulation. They suggest that inflammatory mediators orchestrate the systemic immune response and do not directly contribute to ROS-mediated liver injury. MitoTEMPO affects the redox status of liver cells, which may contribute to its beneficial effect, but still is not sufficient to reduce systemic ROS generation by immune cells. The latter is likely because most of MitoTEMPO is taken up by tissues and only a small fraction by immune cells. This may be explained by the volume/abortive surface area of the tissues, which is much larger than the surface area of the immune cells. This well-conducted study and well-written manuscript should be published after minor issues commented by the reviewer are resolved and contribute interesting findings to the scientific community and research field.

Authors: We thank the Reviewer for the positive evaluation of our study.

Reviewer#1: materials and methods: personally the reviewer would prefer for optimal transparency, to present here the animal experimentation ethic permit number. line 88/89

Authors: We thank the Reviewer for this suggestion. We added the animal experimentation ethic permit number to the Material and Methods section.

The following text was added to the Material and Methods section:

(no. M58003956-2011-9, no. 815758/2014/16)

Reviewer#1: Result section: Figure 1 b-d are missing in my download of the manuscript.

Authors: We are sorry about that. In the original Word file, Figure 1 b-d was present. Most probably, this must have happened during the conversion from the Word file to the PDF file.

Reviewer#1: Figure 1a, “blood and organ sampling” together is a bit misleading, organs were harvested at the end of this period. Of course the reviewer understands what is meant, but maybe this could be optimized.

Authors: Thank you for this suggestion. We changed the text in the Material and Methods and the scheme accordingly.

The text in the Material and Methods section was changed as follows:

Blood samples were taken at 2, 4, 8 and 16 hours after the LPS treatment, liver samples 16 hours after LPS treatment.

Reviewer#1: Results 3.4. line 186 and 187: the reviewer is convinced that in field of the authors everyone knows what EPR technique might be, but maybe not for everyone in general this is clear.

Authors: Thank you for this point. Indeed EPR technique is complex and needed to be explained in more details.

We have added the following text to the Methods section:

EPR is the most appropriate method to detect redox active centers directly in untreated tissues. It can be used to determine redox-active iron-sulfur species (Stich, PMID: 34292554), copper containing compounds (Jakubowska, PMID: 32464500), free radicals and ferrous ions (Kozlov, PMID: 1321074) as well as nitric oxide (Dumitrescu, PMID: 27368066). A particular advantage of this method is that the analytic procedure can be performed directly in frozen tissue biopsies at liquid nitrogen temperature (Dumitrescu, PMID: 27368066). This ensures the absence of artefacts due to processing of tissues, such as homogenization, extraction etc.

Reviewer#1: Discussion, Line 202, “murine model” would be a mouse model, in the manuscript a “rat model” or “rodent model” is described.

Authors: Thank you for this hint. We corrected “murine model” to “rodent model”.

Reviewer 2 Report

1) I did not see a link to figure 1a

2) Panels b-d in Figure 1 did not open on any of my computers. Please check Figure 1.

3) In vivo, in vitro should be written in italics

4) It would be good to provide N-acetyl-3,7-dihydroxy phenoxazine and MitoTracker cell loading images to demonstrate the mitochondrial localization of the probe

Author Response

Point-by-point Response to Reviewer's Comments

Reviewer#2

Reviewer#2: 1) I did not see a link to figure 1a.

Authors: We thank the Reviewer for this hint. We added “Fig. 1a” to the Result Section.

Reviewer#2: 2) Panels b-d in Figure 1 did not open on any of my computers. Please check Figure 1.

Authors:  We are sorry about that. In the original Word file, Figure 1 b-d was present. Most probably, this must have happened during the conversion from the Word file to the PDF file.

Reviewer#2: 3) In vivo, in vitro should be written in italics.

Authors:  Thank you for this suggestion. We corrected in vivo and in vitro accordingly.

Reviewer#2: 4) It would be good to provide N-acetyl-3,7-dihydroxy phenoxazine and MitoTracker cell loading images to demonstrate the mitochondrial localization of the probe.

Authors:  We measured N-acetyl-3,7-dihydroxy phenoxazine with a plate reader, not microscopy. MitoTracker was not used in this study. However, we measured mitochondrial function (please see new additional Figure 1g).

The following text was added to the Methods section:

Measurement of mitochondrial respiration        
Respiratory parameters of mitochondria were monitored using high resolution respirometry (Oxygraph-2k, Oroboros Instruments, Austria) as previously described [9]. Rat liver homogenate was incubated in buffer containing 105 mM KCl , 5 mM KH2PO4, 20 mM Tris-HCl, 0.5 mM EDTA and 5 mg/mL fatty acid-free bovine serum albumin (pH 7.2, 37 °C). Respiration was stimulated by the addition of 5 mM glutamate and 5 mM malate. Transition to State-3 respiration was induced by the addition of 1 mM adenosine di-phosphate. Oxygen consumption rates were obtained by calculating the negative time derivative of the measured oxygen concentration. Respiratory control ratio was calculated by dividing State 3 respiration by State 2 respiration.

The following text was added to the Results section:

We also found that mitochondrial function in the liver is affected by LPS challenge (Fig. 1g). As with the AST release, addition of mitoTEMPO normalized the respiratory control ratio (Fig. 1g).

The following text was added to the Discussion section:

We also observed that respiratory control ratio is increased in response to LPS and normalized upon addition of mitoTEMPO. It has already been reported that upon inflammatory conditions the mitochondrial respiratory function can respond not only by a decrease, but also by an increase in the capacity of electron transport chain (Jeger, PMID: 23496374). Also in our previous studies on rodents, we observed an increase in the State 3 respiration (Kozlov, PMID: 16474010) as we show here while in experiments with peritonitis in pigs the respiratory activity was decreased (Kozlov, PMID: 20180005). The reason why mitochondrial function is upregulated in some cases and downregulated in other cases is not completely clear. However, the normalization of mitochondrial function to its control values by mitoTEMPO suggests that the changes in mitochondrial function observed here reflect pathological changes in the liver and these changes are mediated by mitoROS.

Reviewer 3 Report

In this manuscript, the effects of mitochondrial antioxidants are evaluated in a rat model of endotoxemia. While the investigators have some interesting findings relating to liver damage and distribution of nitric oxide, these findings are very preliminary in nature and not validated by additional measures. Furthermore, there are missing methodological details and the conclusions do not fully match the data. Specific comments are detailed below:

Major

1.    Figure 1. Panels b-d are not present.

2.    Further experimental measures should be used to characterize the effects of mitotempol on liver damage, not just inflammatory markers and AST.

3.    The authors state in figure 3 that the effects of mitotempol disappear during stimulation with PMA however, report a significant effect (although less in magnitude) in bronchoalveolar lavage cells.

4.    Details for EPR measures are missing from the methods.

5.    The authors suggest in the text that mitotempol abolishes differences in free radical production (line 191) however this is not what is indicated in figure 4c.

6.    The authors suggest changes in blood borne NO with their EPR data. This should be verified with other methods.

7.    The authors state that PMNs (not defined) are the primary cell type sampled from the various body compartments. The validation of this should be further clarified in the methods linking it to the supplementary figure.

8.    The authors provide little insight into the endotoxemia literature including fewer than 20 references.

Author Response

Point-by-point Response to Reviewer's Comments

Reviewer#3

Reviewer#3: In this manuscript, the effects of mitochondrial antioxidants are evaluated in a rat model of endotoxemia. While the investigators have some interesting findings relating to liver damage and distribution of nitric oxide, these findings are very preliminary in nature and not validated by additional measures. Furthermore, there are missing methodological details and the conclusions do not fully match the data. Specific comments are detailed below:

Authors:  We thank the Reviewer for his/her constructive criticism. We tried to improve the manuscript and also added additional data (see below).

Reviewer#3: Figure 1. Panels b-d are not present.

Authors:  We are sorry about that. In the original Word file, Figure 1 b-d was present. Most probably, this must have happened during the conversion from the Word file to the PDF file.

Reviewer#3: Further experimental measures should be used to characterize the effects of mitotempol on liver damage, not just inflammatory markers and AST.

Authors:  We thank the Reviewer for this suggestion. We added additional data of mitochondrial respiration measurements to the Results section (new Fig. 1g).

The following text was added to the Methods section:

Measurement of mitochondrial respiration        
Respiratory parameters of mitochondria were monitored using high resolution respirometry (Oxygraph-2k, Oroboros Instruments, Austria) as previously described [9]. Rat liver homogenate was incubated in buffer containing 105 mM KCl , 5 mM KH2PO4, 20 mM Tris-HCl, 0.5 mM EDTA and 5 mg/mL fatty acid-free bovine serum albumin (pH 7.2, 37 °C). Respiration was stimulated by the addition of 5 mM glutamate and 5 mM malate. Transition to State-3 respiration was induced by the addition of 1 mM adenosine di-phosphate. Oxygen consumption rates were obtained by calculating the negative time derivative of the measured oxygen concentration. Respiratory control ratio was calculated by dividing State 3 respiration by State 2 respiration.

The following text was added to the Results section:

We also found that mitochondrial function in the liver is affected by LPS challenge (Fig. 1g). As with the AST release, addition of mitoTEMPO normalized the respiratory control ratio (Fig. 1g).

The following text was added to the Discussion section:

We also observed that respiratory control ratio is increased in response to LPS and normalized upon addition of mitoTEMPO. It has already been reported that upon inflammatory conditions the mitochondrial respiratory function can respond not only by a decrease, but also by an increase in the capacity of electron transport chain (Jeger, PMID: 23496374). Also in our previous studies on rodents, we observed an increase in the State 3 respiration (Kozlov, PMID: 16474010) as we show here while in experiments with peritonitis in pigs the respiratory activity was decreased (Kozlov, PMID: 20180005). The reason why mitochondrial function is upregulated in some cases and downregulated in other cases is not completely clear. However, the normalization of mitochondrial function to its control values by mitoTEMPO suggests that the changes in mitochondrial function observed here reflect pathological changes in the liver and these changes are mediated by mitoROS.

Reviewer#3: The authors state in figure 3 that the effects of mitotempol disappear during stimulation with PMA however, report a significant effect (although less in magnitude) in bronchoalveolar lavage cells.

Authors:  We thank the Reviewer for this important point. Recently, it has been shown that lung tissue can respond differently to “antioxidant treatment” compared to other tissues such as liver, kidney, and spleen (Powell 2018, PMID: 29428024). In their study, Powell and co-workers showed that treatment with triphenylphosphonium, a mitochondria-targeting carrier molecule for antioxidants, increased antioxidant activity in the liver, kidney, and spleen, however, decreased antioxidant activity in the lung. In our recent study, we also observed a difference between the response of lung and other organs to antioxidant treatment (Weidinger 2022, PMID: 35204206).

We added the following to the Result section:

Interestingly, this inhibition disappeared when cells were additionally activated ex vivo by PMA with the exception of bronchoalveolar lavage cells (Fig. 3d-f).

Reviewer#3: Details for EPR measures are missing from the methods.

Authors:  Thank you for this hint. We added the EPR measurement to the Methods section. Following the suggestion of another Reviewer, we also added a short paragraph to explain the EPR method.

The following text was added to the Material and Methods section:

Electron paramagnetic resonance spectroscopy
EPR is the most appropriate method to detect redox active centers directly in untreated tissues. It can be used to determine redox-active iron-sulfur species (Stich, PMID: 34292554), copper containing compounds (Jakubowska, PMID: 32464500), free radicals and ferrous ions (Kozlov, PMID: 1321074) as well as nitric oxide (Dumitrescu PMID: 27368066). A particular advantage of this method is that the analytic procedure can be performed directly in frozen tissue biopsies at liquid nitrogen temperature (Dumitrescu PMID: 27368066). This ensures the absence of artefacts due to processing of tissues, such as homogenization, extraction etc.
EPR spectra were recorded at liquid nitrogen temperature (-196 °C) with a Magnettech MiniScope MS 200 EPR spectrometer (Magnettech Ltd, Berlin, Germany) at modulation frequency 100 kHz, and microwave frequency 9.429 GHz. Settings for NO-Hb spectra in blood samples: microwave power 8.3 mW, modulation amplitude 5 G. NO-Hb complexes were recorded at 3300±200 G and quantified by double integrating the EPR spectra. Settings for NO-Hb and NO-Fe complex in liver samples: microwave power 30 mW, modulation amplitude 6 G. Liver spectra were recorded at 3200±500 G. Settings for p450 (g=2.25), free radicals (g=2.002) and succinate dehydrogenase (g=1.94): microwave power 1 mW, modulation amplitude 5G). Intensities of signals were recorded at 3200 ± 1000 G.  

Reviewer#3: The authors suggest in the text that mitotempol abolishes differences in free radical production (line 191) however this is not what is indicated in figure 4c.

Authors:  Thank you for highlighting this mistake. We changed the wording from “abolished “ to “diminished”.

Reviewer#3: The authors suggest changes in blood borne NO with their EPR data. This should be verified with other methods.

Authors:  Thank you for this suggestion. We also measured NOx in plasma samples using ozone-chemiluminescence technology (Sievers 280i NO Analyzer, General Electrics, USA). We found increased NOx levels after LPS treatment, which could be diminished by mitoTEMPO treatment (new Fig. 4h).

The following text was added to Results section:

In addition, NOx levels in plasma after LPS treatment, determined by ozone-chemiluminescence technology, were increased (Fig. 4h). This increase could be reduced by mitoTEMPO treatment (Fig. 4h).

The following text was added to the Methods section:

Analysis of total nitric oxide       
Total nitric oxide levels (NOx) were analyzed with Sievers 280i-NO Analyzer (General Electrics, USA) as previously described (Pelletier 2006). Plasma samples were injected through a septum into the glass vessel, where NO species were converted by a redox active reagent (VCl3) to NO(g). Subsequent reaction with ozone causing photon emission was detected as chemiluminescence intensity.

Reviewer#3: The authors state that PMNs (not defined) are the primary cell type sampled from the various body compartments. The validation of this should be further clarified in the methods linking it to the supplementary figure.

Authors:  We thank reviewer for this point.

The following text was added to the Methods section:

Cell counts in blood, and peritoneal and bronchoalveolar fluids showed mainly polymorphonuclear leukocytes (PMN, mainly neutrophil granulocytes, Supplementary Figure S1).

Reviewer#3: The authors provide little insight into the endotoxemia literature including fewer than 20 references.

Authors:  Thank you for this comment. Indeed, we did not pay much attention to the description of this experimental model. We have added several references about that to above comments.

 In addition, we also added the following text to the Introduction:

LPS induces acute systemic inflammatory response in rodents (Juskewitch, PMID: 22067909) and humans (Brooks, PMID: 32299854). This model mirrors certain aspects of septic shock in humans, but the correlation between rodent LPS models and the clinical septic shock is poor (Libert, PMID: 30106873). Nonetheless, this model has provided the majority of mechanistic insights into systemic inflammation (Brooks, PMID: 32299854), such as release of cytokines (Flad, PMID: 8330901), induction of apoptosis (Bannerman, PMID: 12736186), signal transduction (Beutler, PMID: 8699845) and others.

Reviewer 4 Report

The study reported in this manuscript by Weidinger A. and colleagues, entitled “Effect of mitoTEMPO on redox reactions in different body compartments upon endotoxemia in rats” aimed to understand how mitochondrial ROS influence redox reactions in a rat model of endotoxemia. To this end, authors analyzed the generation of ROS and the effects of the mitochondrial-targeted antioxidants in blood, abdominal cavity, bronchoalveolar space and liver tissue of rats injected with lipopolysaccharide (LPS).

Major comments

The addressed topic may be of interest, however, I found many major criticisms that make me skeptical about the overall manuscript.

1.    Manuscript structure and English language lack clarity. Paragraphs are not well linked, making the reading and the interpretation very hard. The introduction, results, discussion sections and figure captions are not well developed.

2.  In the result section, figure 1 b, c and d are missed.

3.  The methodology is not sufficiently described, and lack proper controls:

o   Ctrl+M are missed/not reported in figure 1 (e-f)

o   Ctrl (untreated) and Ctrl treated with M are missed/not reported in figure 2 (a-f), figure 3 (a-f).

o   Ctrl treated with M are missed/not reported in figure 4 (b-g)

4.    The study is mainly descriptive and lacks a main driving hypothesis. 

Author Response

Point-by-point Response to Reviewer's Comments

Reviewer#4

Reviewer#4: The study reported in this manuscript by Weidinger A. and colleagues, entitled “Effect of mitoTEMPO on redox reactions in different body compartments upon endotoxemia in rats” aimed to understand how mitochondrial ROS influence redox reactions in a rat model of endotoxemia. To this end, authors analyzed the generation of ROS and the effects of the mitochondrial-targeted antioxidants in blood, abdominal cavity, bronchoalveolar space and liver tissue of rats injected with lipopolysaccharide (LPS). The addressed topic may be of interest, however, I found many major criticisms that make me skeptical about the overall manuscript.

Authors: We thank the reviewer for detailed and constructive evaluation of our manuscript.

Reviewer#4: 1. Manuscript structure and English language lack clarity. Paragraphs are not well linked, making the reading and the interpretation very hard. The introduction, results, discussion sections and figure captions are not well developed.

Authors: Many thanks for this comment. We have revised the manuscript thoroughly and carefully in order to improve its clarity.

Reviewer#4: 2. In the result section, figure 1 b, c and d are missed.

Authors: We are sorry about that. In the original Word file, Figure 1 b-d was present. Most probably, this must have happened during the conversion from the Word file to the PDF file.

Reviewer#4: 3.  The methodology is not sufficiently described, and lack proper controls:

o   Ctrl+M are missed/not reported in figure 1 (e-f)         
o   Ctrl (untreated) and Ctrl treated with M are missed/not reported in figure 2 (a-f), figure 3 (a-f).
o   Ctrl treated with M are missed/not reported in figure 4 (b-g)

Authors: We thank the Reviewer for this important comment, indeed these controls are missing and below we describe the reasons for this.

With regard to figure 2 (a-f), figure 3 (a-f) we could not detect the signals in control samples, they were under the detection limit.

With regard to figures 1 and 4, indeed we do not have these controls. There are two reasons for that.

  1. We aimed to address the question whether mitoTEMPO improves pathological changes induced by LPS, we did not aim to look at changes in controls.

Based on that we did not use these controls in order to kill less animals.

However, we take well this criticism of the Reviewer and we mention now this issue as a limitation of our study. The following text was added to the Discussion section:

Since we have already investigated the effect of mitoTEMPO on control tissues in previous studies (Weidinger 2015, Weidinger 2022), we did not include this group here. Consequently, the conclusions are limited to comparisons between control and LPS on one hand and LPS to LPS with mitoTEMPO on the other hand.

Reviewer#4: 4. The study is mainly descriptive and lacks a main driving hypothesis.

Authors: We agree that a substantial part of the study is descriptive, but we addressed the question whether the results are similar in different compartments. This has an important methodological impact and gives rise to further mechanistic studies. We will definitely employ mechanistic approaches in this model in the future. We hope that the r

Round 2

Reviewer 3 Report

In this revised manuscript, the effects of mitochondrial antioxidants are evaluated in a rat model of endotoxemia. Many improvements have been made by the authors in the revision stage. Methods, results, and overall discussion relating to the field have been markedly improved. However, issues remaining with figure 1 still need to be addressed.

Major

1.    Figure panel 1b is still missing from the revised version of the document, although panels c and d are now included. Also, parts of panel 1a seem to be doubled up.

Author Response

Dear Reviewer,

Many thanks for your comment. We have checked out the issue with the Fig 1b. Surprisingly, in all our machines, we are able to see this panel. Also after conversion of the word file into PDF everything is displayed correctly. It strongly looks like the problem does not come from our side. I submit now both Word and PDF versions of the manuscript with correctly displayed figures and ask the editor to check this issue with technical department of the editorial office.

Sincerely yours.

Andrey Kozlov

On behalf of authors.